# Internalising and externalising behaviour in siblings of children born preterm Preterm birth: Internalising and externalising behaviour of siblings

**Wnurinham Silva**[1*], **Demetris Avraam**[2,3], **Luise Cederkvist**[3], **Johanna Lucia Nader**[4], **Maja Popovic**[5,6], **Hanan El Marroun**[7,8], **Jennifer R. Harris**[9], **Lorenzo Richiardi**[5,6], **Henning Tiemeier**[8,10], **Timothy James Cadman**[11,12], **Julia Jaekel**[13,14,15,16,17,18], **Anne-Marie Nybo Andersen**[3], **Eero Kajantie**[14,19,20,21‡], **Sylvain Sebert**[1‡]

**1** Research Unit of Population Health, Faculty of Medicine, University of Oulu, Oulu, Finland, **2** Department of Public Health, Policy and Systems, University of Liverpool, Liverpool, United Kingdom, **3** Section of Epidemiology, Department of Public Health, University of Copenhagen, Copenhagen, Denmark, **4** Department of Genetics and Bioinformatics, Division of Health Data and Digitalisation, Norwegian Institute of Public Health, Norway, **5** Cancer Epidemiology Unit, Department of Medical Sciences, University of Turin, Turin, Italy, **6** CPO Piemonte, Turin, Italy, **7** Department of Psychology, Education and Child Studies – Erasmus School of Social and Behavioural Sciences, University Rotterdam, Rotterdam, The Netherlands, **8** Department of Child and Adolescent Psychiatry, Erasmus University Medical Center, Rotterdam, The Netherlands, **9** Center for Fertility and Health, The Norwegian Institute of Public Health, Oslo, Norway, **10** Department of Social and Behavioural Sciences, Harvard T.H. Chan School of Public Health, Harvard University, Boston, United States of America, **11** Barcelona Institute for Global Health, Barcelona, Spain, **12** Department of Genetics, Genomics Coordination Center, University Medical Center Groningen, University of Groningen, Groningen, The Netherlands, **13** Psychology, Faculty of Education and Psychology, University of Oulu, Oulu, Finland, **14** Population Health Unit, Finnish Institute for Health and Welfare (THL), Helsinki, Finland, **15** Department of Psychology, University of Warwick, Coventry, United Kingdom, **16** Department of Health Sciences, University of Leicester, Leicester, United Kingdom, **17** Department of Paediatrics I, Essen University Hospital, University of Duisburg-Essen, Duisburg, Germany, **18** Department of Psychology, University of Copenhagen, Copenhagen, Denmark, **19** Clinical Medicine Research Unit, Medical Research Center, Oulu University Hospital, University of Oulu, Oulu, Finland, **20** Department of Clinical and Molecular Medicine, Norwegian University of Science and Technology, Trondheim, Norway, **21** Pediatric Research Centre, Children's Hospital, Helsinki University Hospital and University of Helsinki, Helsinki, Finland

‡ EK and SS are joint senior authors on this work.
* wnurinham.silva@oulu.fi

## Abstract

The family environment in which children grow up is associated with the development of their behaviour. It has been proposed that changes in family dynamics, associated with a child being born preterm, can influence siblings' health. We tested the hypothesis that term-born children (≤ 14 years of age) with younger preterm born siblings are at an increased risk of having higher internalising and externalising behaviour problems than term-born children with term-born siblings. We also compared scores with children without siblings. We used harmonised data from four European birth cohorts. We grouped 65,711 term-born children (49% girls) across the four cohorts as follows: *risk group* (with younger preterm born siblings; n = 427), *reference group* (with only term siblings; n = 12,371), and *only-child group* (without siblings; n = 52,913). We

which permits unrestricted use, distribution, and reproduction in any medium, provided the original author and source are credited.

**Data availability statement:** The data used for this study is third-party data, and without legally permission of distribution or public sharing. Information about data access and governance for this study is explained in detail in a peer-reviewed scientific paper by Prof. Vincent Jaddoe et al (2020): "The LifeCycle Project-EU Child Cohort Network: a federated analysis infrastructure and harmonized data of more than 250,000 children and parents". The principal investigators or home institutions administer permission to the data for external researchers: hence, access to the data is conditional on reasonable request and with approval by each cohort. A description of the data set and third-party sources are displayed online at the EU Child Cohort Variable Catalogue (https://data-catalogue.molgeniscloud.org/catalogue/catalogue/#/). For data request, please find below cohort-specific contact details (email address and/or web). MoBa, Norway Email: jennifer.harris@fhi.no Web: http://www.fhi.no/moba DNBC, Denmark Email: amna@sund.ku.dk Web: https://www.dnbc.dk/access-to-dnbc-data GENR, The Netherlands Email: v.jaddoe@erasmusmc.nl Web: https://generationr.nl/researchers/collaboration/ NINFEA, Italy Email: info@progettoninfea.it, lorenzo.richiardi@unito.it Web: https://www.progettoninfea.it/index_en

**Funding:** The corresponding author of this manuscript (WS) has been funded by the University of Oulu Graduate School (no grant number) and has received financial support from the LifeCycle project in form of a Research Fellowship. EK has received funding from the Academy of Finland, Sigrid Juselius Foundation, Signe and Ane Gyllenberg Foundation and Foundation for Pediatric Research. All cohorts included in this study have received financial support from LifeCycle project. The LifeCycle project received funding from the European Union's Horizon 2020 research and innovation programme (Grant Agreement No. 733206 LifeCycle). In addition, the work of the Norwegian Mother, Father and Child Cohort Study (MoBa) was partly supported by the Research Council of Norway through its Centres of Excellence funding scheme, project number 262700; The Danish National Birth Cohort was established with a significant grant from the Danish National Research Foundation. Additional support was

investigated whether the internalising and externalising z-standardised scores of the term-born children differ by group. The scores came from parent-completed Strengths and Difficulties Questionnaire or Child Behaviour Checklist. Scores of the risk and only-child groups were compared to the reference group. Analyses were conducted in three age groups: ≤ 4, 5–10 and 11–14 years of age. We conducted a two-stage individual participant data meta-analysis and found no evidence of differences in internalising or externalising scores between the risk and the reference groups within any of the age groups. In contrast, the internalising and externalising scores of the only-child participants were higher than the reference group (Internalising- ≤ 4 years: mean difference (MD)=0.06 [95%CI = 0.03,0.08]; 5–10 years: MD = 0.12 [-0.01,0.25]; 11–14 years: MD = 0.07 [0.03,0.12]; Externalising- ≤ 4 years: MD = 0.06 [0.03,0.08]; 5–10 years: MD = 0.10 [0.06,0.15]; 11–14 years: MD = 0.09 [-0.03,0.21]).We found no evidence supporting that having a younger sibling born preterm is a risk factor for increased internalising or externalising behaviour. However, we observed higher internalising and externalising scores in children without siblings compared to those with term-born siblings.

## Introduction

Globally, it is estimated that over 13% of children and adolescents experience mental health disorders [1]. The environment where they grow up is central to their wellbeing and may be associated with the development of mental health disorders [2].

The quality of the home environment may impact the risk of developing internalising [3,4] and externalising problems [5] in childhood and adolescence. Internalising behaviours, characterised by behaviours directed inwards, encompasses conditions where the central feature is disordered mood or emotion, for example stress, depression, or anxiety [6]. Externalising behaviours are directed outwards, and categorized by dysregulated behaviour, such as impulsivity, aggressiveness, disruptiveness, or rule breaking [6]. Both internalising and externalising behaviours in childhood are associated with a range of adult psychiatric disorders and adverse outcomes such as academic failure and functioning in society [7–9].

Families with children born preterm (before 37 weeks of gestation) may experience changes in their home environment stemming from the health care needs of the preterm child. Short- and long-term impacts of preterm birth on parents, such as stress and depression, have been extensively studied [10–12]. For instance, it has been found that family stress levels remain elevated seven years after the birth of a preterm child (11). Such changes in mental health and psychosocial stress may impact parenting, which, in turn, may affect the children's behaviour. Thus, it can be hypothesised that siblings may indirectly be affected by having a preterm born sibling.

Theories of child development and the family environment, such as developmental ecology and family systems theory, posit that children's development occurs in and as a product of interdependent and measurable factors in their proximal family environment [13,14]. According to these theories, aspects such as family dynamics and interactions,

obtained from the Danish Regional Committees, the Pharmacy Foundation, the Egmont Foundation, the March of Dimes Birth Defects Foundation, the Health Foundation and other minor grants. The DNBC Biobank has been supported by the Novo Nordisk Foundation and the Lundbeck Foundation. Follow-up of mothers and children have been supported by the Danish Medical Research Council (SSVF 0646, 271-08-0839/06-066023, O602-01042B, 0602-02738B), the Lundbeck Foundation (195/04, R100-A9193), The Innovation Fund Denmark 0603-00294B (09-067124), the Nordea Foundation (02-2013-2014), Aarhus Ideas (AU R9-A959-13-S804), University of Copenhagen Strategic Grant (IFSV 2012), and the Danish Council for Independent Research (DFF – 4183-00594 and DFF - 4183-00152); Co-authors AMNA and LC, from the Danish National Birth Cohort (DNBC), were supported by EUCAN Connect, grant number 824989; the work of Generation R Study (Gen R) was supported by Stichting Volksbond Rotterdam (to HEM), by the Netherlands Organization for Scientific Research Aspasia (Grant No. 015.016.056, to HEM), by the European Union's Horizon 2020 Research and Innovation Program (HappyMums, Grant Agreement No. 101057390, to HEM), and by the Netherlands Organisation for Health Research and Development (ZonMw Vici project No. 016. VICI.170.200, to HT). The funders had no role in study design, data collection and analysis, decision to publish, or preparation of the manuscript.

**Competing interests:** The authors have declared that no competing interests exist.

**Abbreviations:** ACEs, Adverse childhood experiences; CBCL, Child Behaviour Checklist; DNBC, Danish National Birth Cohort; EUCCN, European Child Cohort Network; GenR, Generation R Study; IPD, Individual participant data; MD, Mean difference; MoBa, Norwegian Mother, Father and Child Cohort Study; NINFEA, Nascita e INFanzia: gli Effetti dell'Ambiente; SDQ, Strengths and Difficulties Questionnaire.

power relations and structure are strong determinants of children's psychological adjustment, with positive familial experiences being associated with better behavioural outcomes [15] and negative experiences associated with more behavioural problems [8,16]. It may be assumed that a preterm born child, on average, requires more support than a term-born child, which influences the resources shared in a family, i.e., fewer resources from parents may be available for siblings of children born preterm.

Two studies investigating the impact of having a preterm born sibling reported no difference in behavioural disorders between term born children with preterm born siblings and those with term-born siblings [17,18]. However, another study found that the group of term-born children with preterm born siblings had more behavioural disorders compared to those with term-born siblings [19]. A systematic review investigating the available evidence on the association between having a preterm born sibling and the risk of poor cognitive function, quality-of-life and mental health of siblings highlighted large methodological heterogeneity and mixed evidence in the existing research [18–22]. In addition, they found no evidence to support or refute an association between having a preterm born sibling and the risk of poor health outcomes [22]. Considering that children with preterm born siblings are potentially at increased risk of experiencing mental health disorders, but grossly overlooked, further studies are needed to clarify this relationship. Such studies could provide invaluable evidence of need of targeted interventions to this group.

The present study investigates the relationship between having a younger preterm born sibling and the risk of increased internalising and externalising problems of the term-born child. We hypothesised that term-born children with a younger preterm sibling are at an increased risk of developing internalising and externalising symptoms during childhood and early adolescence. Our study design also includes a group of children with no siblings in the cohorts. This helps to ascertain whether any observed association or the lack of one is due to specifically having a sibling born preterm or related to simply having a sibling (i.e., independent of their birth outcome). We performed individual participant data (IPD) meta-analysis and pooled regression analysis using a unique source of harmonized data from four European birth cohorts, with internalising and externalising data for 65,711 children from infancy to 14 years of age.

## Materials and methods

### Participating cohorts

The European Child Cohort Network (EUCCN) is a network of birth cohorts from across Europe and Australia, who have generated a comprehensive set of harmonised variables [23]. In 2021, all EUCCN cohorts were invited to participate in the current study. Cohorts were eligible if they had data on siblings (full or maternal half siblings), internalising and externalising measurements, and information on gestational age. Based on these eligibility criteria, four cohorts participated in the study: the Norwegian Mother, Father and Child Cohort Study in Norway (MoBa, n = 105,311) [24], the Danish National Birth Cohort in Denmark (DNBC, n = 96,825) [25], the Generation R Study in the Netherlands (GenR, n = 9,901) [26], and the Nascita e INFanzia: gli Effetti dell'Ambiente in Italy (NINFEA, n = 7,642) [27] (Table 1).

**Table 1. Cohort-specific study characteristics and information on exposure and outcome measurements.**

| Cohort | | MoBa | DNBC | GenR | NINFEA | All cohorts |
|---|---|---|---|---|---|---|
| **Cover Area** | | Norway | Denmark | The Netherlands | Italy | – |
| **Years of recruitment** | | 1999-2008 | 1996-2002 | 2002-2006 | 2005-2016 | – |
| **Timing of Recruitment** | | In pregnancy (17–18 weeks) | In pregnancy (12 weeks) | In pregnancy (<18 weeks) | In pregnancy | – |
| **Source of GA Information** | | Ultra-sound | A combination of last menstrual period and ultra-sound | Ultra-sound (pre-natally); Medical records or self-recorded for partici-pants (postnatally) | Last menstrual period, ultra-sound (and report of GA at birth when first two not available) | – |
| **Source of Internalising and Externalis-ing scores** | | Parent-reported | Parent-reported | Parent-reported | Parent-reported | – |
| **Age of measurements** | ≤4 y | 3-18 months | | 2-48 months | | – |
| | 5-10 y | 5-8 y | 7-8 y | •6 y •10 y | | – |
| | 11-14 y | | 11-14 y | | 13-14 y | – |
| **Nr of children by cohort, n** | | 105,311 | 96,825 | 9,901 | 7,642 | 219,679 |
| **Total term born children in cohort, n (%)** | | 96,096 (91) | 90, 629 (94) | 9,180 (93) | 6,493 (85) | 202,398 (92) |
| **Total preterm born children in cohort, n (%)** | | 6,531 (6) | 6,192 (6) | 480 (5) | 409 (5) | 13,612 (6) |
| **Total GA information missing in cohort, n (%)** | | 2 684 (3) | 4 (<1%) | 241(2) | 740 (10) | 3669 (2) |
| **Instruments used** | | CBCL | SDQ | CBCL | SDQ | – |
| **\* Focal Children, n** | | 41,546 | 41,278 | 4,752 | 4,239 | 91,815 |
| **\* Nr of Focal children by group, n (%)** | *Risk group* | 344 (<1%) | 149 (<1%) | 13 (<1%) | 23 (<1%) | 529 (1%) |
| | *Reference group* | 9,274 (22) | 4,620 (11.2) | 396 (8.3) | 509 (12) | 14,799 (16) |
| | *Only child group* | 31,928 (77) | 36,509 (88) | 4,343 (91) | 3,707 (88) | 76,487 (83) |
| **\*\* Nr of Focal chil-dren by group with at least one internal-ising measurement, n (%)** | *Risk group* | 292 (85) | 107 (72) | 12 (92) | 10 (43) | 421 (80) |
| | *Reference group* | 8,217 (89) | 3,587 (78) | 356 (90) | 151 (30) | 12,311 (83) |
| | *Only child group* | 23,710 (74) | 25,009 (69) | 3,290 (76) | 570 (15) | 52,579 (69) |
| **\*\* Nr of Focal chil-dren by group with at least one external-ising measurement, n (%)** | *Risk group* | 297 (86) | 106 (71) | 12 (92) | 10 (43) | 425 (80) |
| | *Reference group* | 8,219 (89) | 3,587 (78) | 358 (90) | 151 (30) | 12,315 (83) |
| | *Only child group* | 23,705 (74) | 25,002 (68) | 3,308 (76) | 571 (15) | 52,586 (69) |

Focal children: term-born children (≥ 37 weeks GA) for whom who were interested analysing behaviour problems;

*Total number of Focal children after excluding twins and triplets and missing plurality and GA (including those without internalising and externalising behaviour measurements);

** Focal children after excluding twins and triplets and missing plurality, GA, internalising and externalising variables;

Groups: risk group (term-born children with preterm born sibling(s)); reference group (term-born children with term-born sibling(s)); only-child group (term-born children without siblings); Abbreviations: GA: gestational age; MoBa: Norwegian Mother, Father and Child Cohort Study; DNBC: Danish National Birth Cohort; GenR: the Generation R Study; NINFEA: Nascita e INFanzia: gli Effetti dell'Ambiente; SDQ: Strengths and Difficulties Questionnaire CBCL: Child Behavior Checklist.

## Study population

There were 219,679 mother-child dyads across the four participating cohorts, as shown in the study flowchart (Fig 1). A total of 202,398 (92%) children were born at term and 13,612 (6%) were born preterm (Table 1). Data on gestational age at birth were missing for 3,669 (2%) children. Due to challenges associated with including twins in epidemiological studies [28], children from mothers with twins and triplets were excluded. In addition, to guarantee that we had complete data for all included family dyads, children from mothers with missing information on gestational age, or plurality (i.e., twins, singletons) for one or more children were also excluded. Finally, only children with at least one available internalising and externalising behaviour measurement were included in the analysis.

## Assessment of the exposure

Preterm birth was defined as less than 37 completed weeks of gestation according to the first day of the mother's last menstrual period and ultrasound (DNBC, NINFEA) or just from ultrasound (MoBa, GenR) (Table 1). Babies born at 37 weeks and later were classified as term-born.

## Only-child group

To support the test of our hypothesis we also included in our study design a group of children with no siblings in the cohorts. The purpose was to ascertain whether any observed association or the lack of one was due to having a sibling

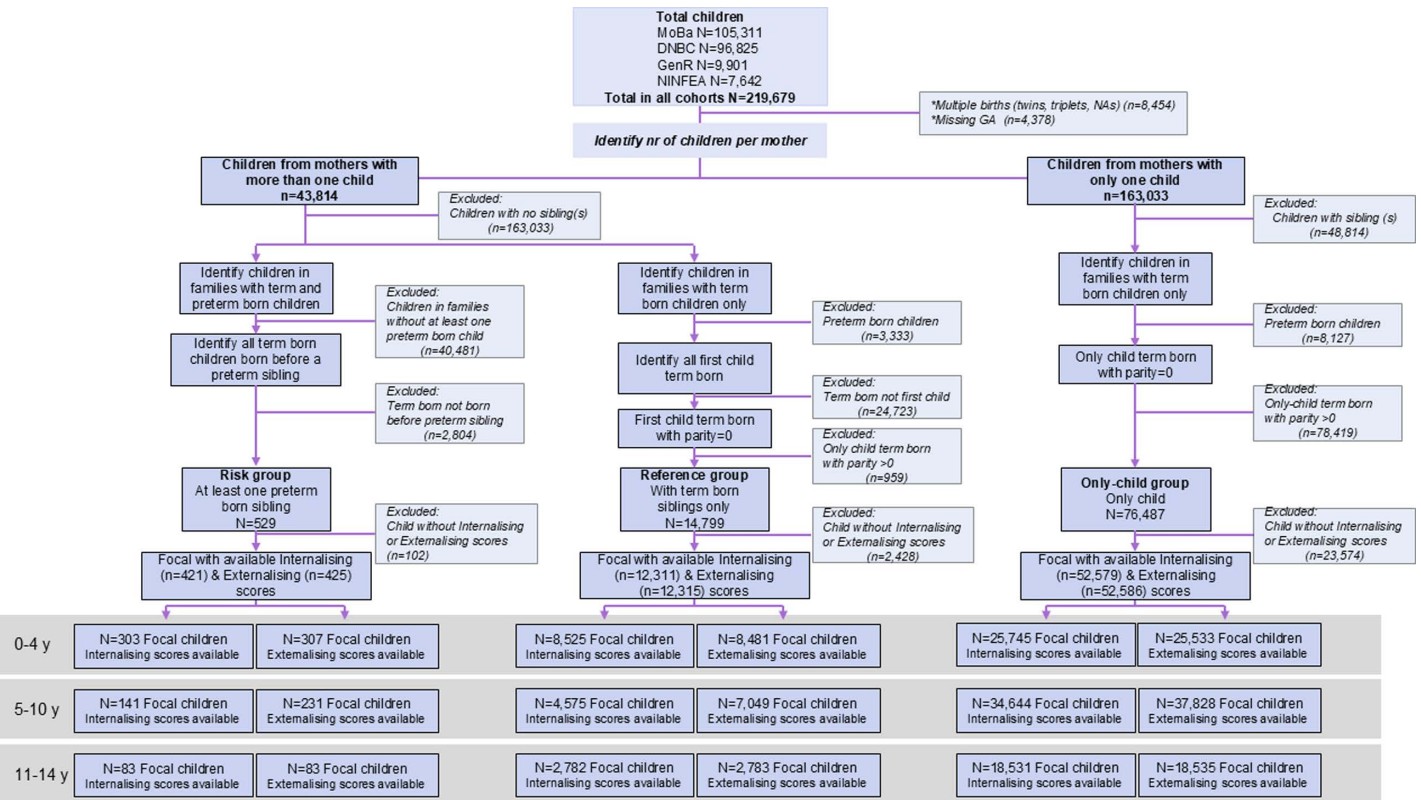

**Fig 1. Study flowchart.** Flowchart illustrating participants included in the study. Only focal children with at least one available internalising and externalising behaviour measurement (grey area) were included in the analyses.

born preterm or simply related to having a sibling (i.e., independent of their birth outcome). The behavioural scores of the group of children with preterm born siblings and children with no siblings were compared against the scores of the term-born children with term-born siblings.

### Definition of focal child

As we were interested in investigating the association between having a younger sibling born preterm and the internalising and externalising behaviour of the term-born children, we classified the children into three groups based on their sibling status (S1 Fig). Henceforth we refer to the term-born children, for whom analysed scores of internalising and externalising behaviours, in all three groups as *focal children,* a designation used elsewhere [29]. The siblings of the focal children, either born preterm or at term, have been simply referred to as *siblings.*

Certain inclusion criteria were specific to each group:

1. Term-born children with preterm born siblings (risk group). The focal children were selected if they were older than and closest in birth position to their preterm sibling. No other restriction was applied to their birth position. Family sizes ranged from two to five children and preterm birth position varied from family to family.

2. Term-born children with term-born siblings (reference group). In this group, in the absence of a specific exposure, such as preterm born sibling, we wanted to ensure that all focal children in this group were firstborns who had younger siblings. Hence, we included only firstborn children whose mothers had null parity, defined as no previous children at the time of the first birth in our dataset, to ensure that there are no previous exposures to circumstances related to siblings.

3. Term-born children without any siblings in the cohort (only-child group). To ensure mothers of the included children did not have any children before entering the cohort, we included only firstborn children whose mothers had null parity at their birth.

### Assessment of the outcomes

Internalising and externalising behaviours were measured using one of two well-established parent-report instruments: the Strengths and Difficulties Questionnaire (SDQ; DNBC and NINFEA) [30], or the Child Behaviour Checklist (CBCL; GenR and MoBa) [31]. The SDQ contains 25-items measured on a 3-point Likert scale ("not true", "somewhat true", "certainly true") containing five subscales: Emotional Problems, Conduct Problems, Hyperactivity, Peer Problems, and Pro-social behaviour. Internalising scores are calculated by combining the Emotional Problems and Peer Problems subscales, while externalising scores are calculated by combining the Conduct Problems and Hyperactivity subscales. The CBCL contains 99-items rated on a 3-point Likert scale ("not true", "sometimes true", "often true"), containing eight subscales: Delinquent Behavior, Aggressive Behavior, Withdrawn, Somatic Complaints, Anxious/Depressed, Social Problems, Thought Problems, and Attention Problems. Internalising scores are calculated by combining the Withdrawn, Somatic Complaints, and Anxious/Depressed subscales, while externalising scores are calculated by combining the Delinquent and Aggressive Behaviours.

To allow statistical comparability across instruments we z-standardized the internalising and externalising scores by age according to the term born children's scores within each cohort before the main analysis. Z-scores have been calculated by subtracting from each individual raw score (internalising or externalising) the mean value and dividing by the standard deviation of the raw scores of all term born children in the cohort. This transformation standardises the scores to a distribution with mean equal to 0 and standard deviation equal to 1. This helps overcome scaling differences across the harmonised dimensional scores within the respective age-bands of the CBCL and SDQ.

Parent reports of children's internalising and externalising behaviour were collected by participating cohorts from early childhood to early adolescence, however, data collection points differed from cohort to cohort (Table 1). We analysed the outcome variables in three age groups created based on data availability and developmental stages: ages equal and less than 4 years (early childhood), ages from 5 to 10 years (mid-childhood), and ages from 11 to 14 years (pre-adolescence

and early adolescence). In each age group, if a child had multiple scores, we used the latest available. Considering that data collection was conducted at different ages in each cohort, the above-mentioned standardisation of the outcomes minimizes scaling differences across cohorts within each age group.

## Confounders

A list of harmonised confounders was selected a priori based on the available evidence on their association with both preterm birth and internalising and externalising behaviour of children [32,33]. These were: maternal age at the birth of the focal child (years), maternal highest educational attainment at birth of the focal child (high, medium, low), maternal smoking during pregnancy (yes, no), maternal alcohol intake during pregnancy (yes, no). Maternal educational level was harmonized across cohorts based on the International Standard Classification of Education 97 (ISCED-97) and consisted of three categories: Low (No education to lower secondary; ISCED-97 categories 0-2), Medium (Upper and post-secondary; ISCED-97 categories 3-4), High (Degree and above; ISCED-97 categories 5-6). Despite evidence of the impact of maternal mental health on internalising and externalising behaviour of children we have not been able to include such variable as a confounder due to a high proportion of missing values. The harmonised data indeed include an indicator for any exposure to relatives such as parents or other individuals close to the child with mental disorders, however the proportion of missing values in this yearly repeated variable ranged from 67.3% in DNBC to 100% in MoBa.

## Statistical analysis

All analyses were conducted through DataSHIELD [34,35]. We specifically used the R-based DataSHIELD packages dsBaseClient (version 6.3.0) and dsHelper (version 0.4.21).

We tested the association between having a sibling born preterm and the internalising and externalising behaviour of the focal children using descriptive statistics and linear regression analysis. Descriptives were conducted for each cohort individually and for all cohorts combined. We conducted a two-stage individual participant data (IPD) meta-analysis. With this approach, a regression model is fitted in each cohort separately and then the cohort-specific estimates and standard errors are combined with random effects meta-analysis. We used the rma function from the metafor R package (version 4.6-0) with the Restricted Estimate Maximum Likelihood method for the random-effects meta-analysis. With this method, the combined estimate is given as the weighted average of the cohort-specific estimates where the weights are defined as the inverse of the variance of the estimates. Between-cohort heterogeneity was evaluated with two-stage IPD meta-analysis using I-squared ($I^2$) and chi-squared Q-statistics. The analysis was conducted by age groups (≤ 4 years, 5–10 years, 11–14 years). We performed complete case analysis, including participants with at least one internalising and externalising score available in each cohort. Unadjusted models and models adjusted for confounders were used. The regression estimates can be interpreted as the mean difference in internalising and externalising z-scores for a group taking the Reference group as a reference. Cohort specific estimates in separate age groups are provided as supporting information (S2 Table). Finally, we conducted sensitivity analyses to explore whether associations remained, by applying (i) one-stage IPD analysis adjusting the models by cohort indicators, and (S3 Table) (ii) "across all ages" analyses using individual average internalising and externalising scores from age 0–14 years (S2 Fig). The one-stage IPD analysis is designed to give similar results as a pooled analysis where all the data could be combined and analysed jointly. DataSHIELD allows this method in an iteration process where no individual-level information is transferred or shared between the participating cohorts, but only non-disclosive summary statistics (i.e., score vectors and information matrices) (The procedure is described in [35]).

## Ethics approval

Once eligibility to participate in this study was confirmed, the lead author obtained a waiver of consent (data access or data transfer agreement) from each cohort. DataSHIELD is a federated analysis platform which means that rather than

sharing or transferring individual-level data, only non-disclosive low-dimensional summary statistics are made accessible upon request. Through this platform, harmonized and pseudonymised data is shared in compliance with both European and national data protection, patient's rights, and research ethics regulation. Each participating cohort obtained written consent from their study participants and written informed consent was obtained from the parent/guardian of each participant under 18 years of age (S1 Text). All cohorts obtained ethical approval. MoBA: ethical approval was obtained from the Regional Committees for Medical and Health Research Ethics (REK Sør-Øst C: 2018/427); DNBC: the cohort obtained ethical approval from the regional scientific ethical Committee for the municipalities of Copenhagen and Frederiksberg (Ref. no (KF) 01-471/94); GenR: ethical approval was obtained from the Medical Ethical Committee of the Erasmus Medical Center, Rotterdam. The reference numbers of the ethical documents are as follows: phase 1 (fetal period) MEC 198.782/2001/31; phase 2 (0–4 years) MEC 217.595/2002/202; phase 3 (6 and 10 years) MEC-2007-413; MEC-2010-084; MEC-2012-165; phase 4 (13 and 17 years) MEC 2015-749); NINFEA: the Ethical Committee of the San Giovanni Battista Hospital and CTO/CRF/Maria Adelaide Hospital of Turin approved the NINFEA study (approval N.0048362, and subsequent amendments).

## Results

### Descriptive statistics

From the total of 91,815 focal children in the risk, reference and only-child groups, 65,711 (72%) had at least one internalising and/or externalising behavioural score (Table 1). Descriptive information for complete cases in each cohort and for the cohorts combined is provided in Table 2. The mean internalising and externalising z-scores of the children included in the analysis are provided in Tables 3 and 4. Descriptives including children without internalising and externalising scores is provided in supplementary materials (S1 Table).

In total, there were 427, 12,371 and 52,913 focal children with at least one internalising and/or externalising score in the risk, reference, and only-child groups, respectively. The characteristics of the three groups are provided in Table 2. The characteristics of the children with complete available scores did not differ significantly from the overall population including children without available internalising and externalising scores (S1 Table).

### Focal children: Internalising behaviour

The results showed no evidence for an association between having a sibling born preterm and higher internalising behavioural scores at ages ≤4, 5–10 or 11–14 years (Fig 2). For the only-child group, the two-stage meta-analysis showed higher scores in the only-child group at ages ≤4 (MD)=0.06 [95%CI = 0.03, 0.08]) and 11–14 (MD = 0.07 [0.03, 0.12]) years when compared to the reference group (Fig 2). Between study heterogeneity, in the risk group, was low at age ≤ 4 and 11–14 ($I^2$ = 0.0%, 8.4%, respectively) and considerable at 5–10 ($I^2$ = 62.9%). For the only-child group it was also low at age ≤ 4 and 11–14 ($I^2$ = 0.0%) and considerable at 5–10 years of age ($I^2$ = 89.9%).

These results are consistent with the results of the "across all ages" meta-analysis, which showed no evidence of association for the risk group but showed higher mean internalising scores for the only-child group compared to the reference group (S2 Fig).

### Focal children: Externalising behaviour scores

We observed no difference in externalising scores between the risk and reference groups at ≤4, 5–10 or 11–14 years of age (Fig 3). In the only child-group externalising scores were above the mean of the reference group at ages ≤4 (MD = 0.06 [0.03, 0.08]) and 5–10 MD = 0.10 [0.06, 0.15] years. Between study heterogeneity, in the risk group, was low in all age groups ($I^2$ = 0.0%). In the only-child group it was low at ages ≤4 ($I^2$ = 0.0%) and moderate at ages 5–10 and 11–14 ($I^2$ = 51.8%, 52.8%, respectively).

Table 2. Characteristics of the focal children in the risk group, reference group and only-child group (complete cases).

| Cohort | MoBa (n=32,545) | | | DNBC (n=28,706) | | | GenR (n=3,728) | | |
|---|---|---|---|---|---|---|---|---|---|
| Variables | Risk group (n=298) | Reference group (n=8,274) | Only-child group (n=23,973) | Risk group (n=107) | Reference group (n=3,587) | Only-child group (n=25,012) | Risk group (n=12) | Reference group (n=359) | Only-child group (n=3,357) |
| Male | 146 (49) | 4,240 (51) | 12,254 (51) | 54 (50) | 1,808 (50) | 12,645 (51) | 4 (33) | 191 (53) | 1,648 (49) |
| Female | 152 (51) | 4,034 (49) | 11,719 (49) | 53 (50) | 1,779 (50) | 12,367 (49) | 8 (67) | 168 (47) | 1,709 (51) |
| Focal GA, weeks: mean (SD) | 39.5 (1.3) | 40.2 (1.3) | 40.2 (1.3) | 39.6 (1.5) | 40.30 (1.3) | 40.2 (1.3) | 40.05 (1.6) | 40.7 (1.4) | 40.7 (1.4) |
| Sibling GA, weeks: mean | 34.63 | 40.17 | – | 34.75 | 40.17 | – | 33.98 | 40.70 | – |
| Maternal age, years: mean (SD) | 28.9 (4.2) | 28.3 (3.8) | 28.8 (4.5) | 29.2 (4.2) | 28.04 (3.5) | 28.4 (3.9) | 31.6 (3.7) | 30.8 (4.2) | 29.8 (5.1) |
| Maternal education: n (%) | | | | | | | | | |
| High | 199 (67) | 5,727 (69) | 14,996 (63) | 43 (40) | 1,652 (46) | 9,534 (38) | 11 (92) | 242 (67.4) | 1,541 (46) |
| Medium | 82 (28) | 1,974 (24) | 6,965 (29) | 22 (21) | 966 (27) | 6,442 (26) | Not shown* | 107 (29.8) | 1,455 (43) |
| Low | 5 (2) | 92 (1) | 506 (2) | 14 (13) | 333 (9) | 2,538 (10) | – | 9 (2.5) | 176 (5) |
| NA | 12 (4) | 481 (6) | 1,506 (6) | 28 (26) | 636 (18) | 6,498 (26) | – | 1 (0.3) | 185 (6) |
| Pregnancy smoking: n (%) | | | | | | | | | |
| No | 217 (73) | 6,486 (78) | 17,871 (75) | 81 (76) | 2,771 (77) | 18,171 (73) | 7 (58) | 272 (76) | 2,195 (65) |
| Yes | 81 (27) | 1,788 (22) | 6,102 (25) | 25 (23) | 763 (21) | 6,625 (26) | 5 (42) | 58 (16) | 831 (25) |
| NA | – | – | – | 1 (<1%) | 53 (1) | 216 (1) | – | 29 (8) | 331 (10) |
| Pregnancy alcohol intake: n (%) | | | | | | | | | |
| No | 202 (68) | 5,222 (63) | 16,082 (67) | 40 (37) | 2,083 (58) | 10,928 (44) | Not shown* | 86 (24) | 1,259 (38) |
| Yes | 95 (32) | 3,047 (37) | 7,826 (33) | 66 (62) | 1,442 (40) | 13,798 (55) | 9 (75) | 238 (66) | 1,700 (51) |
| NA | 1 (<1%) | 5 (<1%) | 65 (<1%) | 1 (1) | 62 (2) | 286 (1) | 1 (8) | 35 (10) | 398 (12) |

| Cohort | NINFEA (n=732) | | | All cohorts (n=65,711) | | |
|---|---|---|---|---|---|---|
| Variables | Risk group (n=10) | Reference group (n=151) | Only-child group (n=571) | Risk group (n=427) | Reference group (n=12,371) | Only-child group (n=52,913) |
| Male | 3 (30) | 87 (58) | 277 (49) | 207 (48) | 6,326 (51) | 26,824 (51) |
| Female | 7 (70) | 64 (42) | 294 (51) | 220 (52) | 6,045 (49) | 26,089 (49) |
| Focal GA, weeks: mean (SD) | 39.52 (0.8) | 39.97 (1.3) | 39.89 (1.3) | 39.58 (1.4) | 40.3 (1.3) | 40.26 (1.3) |
| Sibling GA, weeks: mean (SD) | 35.42 | 39.84 | – | 34.8 | 40.18 | – |
| Maternal age, years: mean (SD) | 31.30 (1.6) | 31.76 (3.5) | 32.96 (3.8) | 29.14 (4.2) | 28.36 (3.8) | 28.73 (4.3) |
| Maternal education: n (%) | | | | | | |
| High | 6 (60) | 94 (62) | 357 (63) | 270 (63) | 7,948 (64) | 29,476 (56) |
| Medium | 4 (40) | 51 (34) | 195 (34) | 107 (25) | 2,823 (23) | 13,437 (25) |
| Low | – | 5 (3) | 18 (3) | 27 (6) | 835 (7) | 6,218 (12) |
| NA | – | 1 (1) | 1 (<1%) | 23 (5) | 765 (6) | 3,782 (7) |
| Pregnancy smoking: n (%) | | | | | | |
| No | 10 (100) | 137 (91) | 527 (92) | 315 (74) | 9,666 (78) | 38,764 (73) |
| Yes | – | 14 (9) | 43 (8) | 111 (26) | 2,623 (21) | 13,601 (26) |

*(Continued)*

Table 2. (Continued)

| Cohort | MoBa (n=32,545) | | | DNBC (n=28,706) | | | GenR (n=3,728) | | |
|---|---|---|---|---|---|---|---|---|---|
| **Variables** | **Risk group (n=298)** | **Reference group (n=8,274)** | **Only-child group (n=23,973)** | **Risk group (n=107)** | **Reference group (n=3,587)** | **Only-child group (n=25,012)** | **Risk group (n=12)** | **Reference group (n=359)** | **Only-child group (n=3,357)** |
| NA | – | – | | 1 (<1%) | 1 (<1%) | | 82 (1) | | 548 (1) |
| **Pregnancy alcohol intake: n (%)** | | | | | | | | | |
| No | 10 (100) | 82 (54) | 262 (46) | 298 (52) | 254 (59) | | 6,832 (55) | | 28,567 (54) |
| Yes | | 63 (42) | | 170 (40) | | 5,431 (44) | | | 23,586 (45) |
| NA | | 6 (4) | 11 (2) | 3 (1) | | 108 (1) | | | 760 (1) |

Focal children: term-born children (≥ 37 weeks GA) for whom who were interested analysing behaviour problems. Table includes total number of Focal children by group after all exclusions criteria (complete cases).

Groups: risk group (term-born children with preterm born sibling(s)); reference group (term-born children with term-born sibling(s)); only-child group (term-born children without siblings);

*Fewer than 3 in the summary statistic so value has not been presented owing to risk of disclosure;

Abbreviations: GA: gestational age; MoBa: Norwegian Mother, Father and Child Cohort Study in Norway; DNBC: Danish National Birth Cohort in Denmark; GenR: the Generation R Study; NINFEA: Nascita e INFanzia: gli Effetti dell'Ambiente.

**Table 3. Focal children with available internalising behaviour measurements.**

| Internalising scores | Group | Age groups | Mean z-score (SD) | No of Focal children with available scores (by age group and group), n | No of Focal children with available scores (by group), n |
|---|---|---|---|---|---|
| MoBa (Norway, 1999–2008) | Risk group | ≤4 | 0.027 (1.021) | 291 | 292 |
| | | 5-10 | 0.468 (1.125) | 42 | |
| | | 11-14 | – | – | |
| | Reference group | ≤4 | 0.088 (0.998) | 8,198 | 8,217 |
| | | 5-10 | 0.056 (1.007) | 1,196 | |
| | | 11-14 | – | – | – |
| | Only-child group | ≤4 | 0.156 (1.047) | 23,202 | 23,710 |
| | | 5-10 | 0.172 (1.035) | 10,288 | |
| | | 11-14 | – | – | – |
| DNBC (Denmark, 1996–2002) | Risk group | ≤4 | – | – | 107 |
| | | 5-10 | -0.015 (0.791) | 91 | |
| | | 11-14 | -0.184 (0.829) | 73 | |
| | Reference group | ≤4 | – | – | 3,587 |
| | | 5-10 | 0.099 (1.014) | 3,052 | |
| | | 11-14 | -0.020 (0.974) | 2,627 | |
| | Only-child group | ≤4 | – | – | 25,009 |
| | | 5-10 | 0.130 (1.036) | 21,430 | |
| | | 11-14 | 0.064 (1.017) | 17,944 | |
| GenR (The Netherlands, 2002–2006) | Risk group | ≤4 | -0.231 (0.815) | 12 | 12 |
| | | 5-10 | 0.215 (0.637) | 8 | |
| | | 11-14 | – | – | |
| | Reference group | ≤4 | 0.017 (0.968) | 327 | 356 |
| | | 5-10 | -0.144 (0.805) | 327 | |
| | | 11-14 | – | – | – |
| | Only-child group | ≤4 | 0.078 (1.005) | 2,534 | 3,290 |
| | | 5-10 | 0.110 (1.085) | 2,926 | |
| | | 11-14 | – | – | – |
| NINFEA (Italy, 2005–2016) | Risk group | ≤4 | – | – | 10 |
| | | 5-10 | – | – | |
| | | 11-14 | -0.519 (0.525) | 10 | |
| | Reference group | ≤4 | – | – | 151 |
| | | 5-10 | – | – | – |
| | | 11-14 | 0.039 (0.928) | 151 | |
| | Only-child group | ≤4 | – | – | 570 |
| | | 5-10 | – | – | – |
| | | 11-14 | 0.020 (1.032) | 570 | |
| All cohorts | Risk group | ≤4 | 0.0172 (1.014) | 303 | 421 |
| | | 5-10 | 0.141 (0.897) | 141 | |
| | | 11-14 | -0.224 (0.798) | 83 | |
| | Reference group | ≤4 | 0.085 (0.997) | 8,525 | 12,311 |
| | | 5-10 | 0.071 (0.999) | 4,575 | |
| | | 11-14 | -0.016 (0.972) | 2,782 | |
| | Only-child group | ≤4 | 0.1490 (1.043) | 25,745 | 52,579 |
| | | 5-10 | 0.141 (1.040) | 34,644 | |
| | | 11-14 | 0.063 (1.017) | 18,531 | |

*(Continued)*

**Table 3.** (Continued)

| Internalising scores | Group | Age groups | Mean z-score (SD) | No of Focal children with available scores (by age group and group), n | No of Focal children with available scores (by group), n |
|---|---|---|---|---|---|

Number of focal children with available internalising behaviour measurements, by group and by age group;

Focal children: term-born children (≥ 37 weeks GA) for whom we are analysing scores of internalising and externalising problems;

Groups: risk group (term-born children with preterm born sibling(s)); reference group (term-born children with term-born sibling(s)); only-child group (term-born children without siblings);

Years given in parentheses are years off recruitment, in respective cohort;

In each age group, if a child had multiple scores, we used the latest available;

Abbreviations: MoBa: Norwegian Mother, Father and Child Cohort Study in Norway; DNBC: Danish National Birth Cohort in Denmark; GenR: the Generation R Study; NINFEA: Nascita e INFanzia: gli Effetti dell'Ambiente.

These results are consistent with the results of the "across all ages" meta-analysis, which showed no evidence of association for the risk group but showed higher mean externalising scores for the only-child group compared to the reference group (S2 Fig).

## Discussion

We hypothesized that term-born children with preterm born siblings would have higher internalising and externalising behaviour scores, compared with term-born children with term-born siblings. To test our hypothesis, we conducted a two-stage IPD meta-analysis using data from 65,711 focal children with at least one available internalising and/or externalising scores, and their siblings, from four European birth cohorts.

We found no evidence that children with a younger preterm born sibling have lower or higher internalising or externalising behavioural scores compared to children with only term-born siblings. In line with our findings, a study conducted in the USA (N = 85) investigating mental health outcomes amongst children aged between 1.5 and 5 years reported no evidence of association between having a younger preterm born sibling and internalising and externalising problems [17]. However, that study suggested that term-born children with term-born siblings exhibited higher anxiety and depression scores than children with preterm siblings [17]. A study on a sample of adult preterm born siblings (N = 173) in Canada found no evidence of difference in reported anxiety and mood disorder between the group of adults with preterm siblings and the group with term-born siblings [18]. However, the effects were derived from responses by the preterm adults reporting on their siblings' symptoms. This design may carry important technical (i.e., selection, recall) and emotional biases, in addition to being underpowered. Future studies would benefit from collecting data directly from the siblings themselves.

Our results indicate that children who are an only-child in the cohort nearly consistently score higher on internalising and externalising behaviours when compared to the reference group. Sensitivity analysis of internalising and externalising behaviour with no age groups, showed the same trend. Our results are consistent with the results of a study conducted in China investigating internalising and externalising behaviour of 6–12 years old only-children and firstborns whose mothers were pregnant with a second child. That study showed that only-children had higher internalising and externalising scores compared to their peers with siblings [36]. Although generalizability of that study is limited due to the specific context of the Chinese population and family planning policies, similar findings have also been reported in another study conducted in the USA [37]. This study evaluating self-control, interpersonal skills and externalising behaviour between only-children and children with one or more siblings, suggested that children with one or more siblings fared better than their only-child peers in all domains. This study reported that factors such as sex of siblings or birth order had little effect on the

**Table 4. Focal children with available externalising behaviour measurements.**

| Externalising scores | Group | Age groups | Mean z-score (SD) | No of Focal children with available scores (by age group and group), n | No of Focal children with available scores (by group), n |
|---|---|---|---|---|---|
| MoBa (Norway, 1999–2008) | Risk group | ≤4 | 0.053 (0.980) | 295 | 297 |
| | | 5-10 | -0.124 (0.964) | 133 | |
| | | 11-14 | – | – | |
| | Reference group | ≤4 | 0.0003 (0.976) | 8,155 | 8,219 |
| | | 5-10 | -0.022 (0.969) | | 3,671 |
| | | 11-14 | – | | – |
| | Only-child group | ≤4 | 0.069 (1.001) | 22,978 | 23,705 |
| | | 5-10 | 0.106 (1.011) | | 13,457 |
| | | 11-14 | – | | – |
| DNBC (Denmark, 1996–2002) | Risk group | ≤4 | – | – | 106 |
| | | 5-10 | -0.156 (0.880) | 90 | |
| | | 11-14 | -0.382 (0.851) | 73 | |
| | Reference group | ≤4 | – | – | 3,587 |
| | | 5-10 | -0.080 (0.935) | | 3,052 |
| | | 11-14 | -0.190 (0.928) | | 2,627 |
| | Only-child group | ≤4 | – | – | 25,002 |
| | | 5-10 | 0.015 (0.999) | | 21,419 |
| | | 11-14 | -0.030 (0.988) | | 17,944 |
| GenR (The Netherlands, 2002–2006) | Risk group | ≤4 | -0.336 (0.660) | 12 | 12 |
| | | 5-10 | 0.512 (2.151) | 8 | |
| | | 11-14 | – | – | |
| | Reference group | ≤4 | -0.036 (0.975) | 326 | 358 |
| | | 5-10 | -0.116 (0.880) | | 326 |
| | | 11-14 | – | | – |
| | Only-child group | ≤4 | 0.073 (1.019) | 2,555 | 3,308 |
| | | 5-10 | 0.076 (1.039) | | 2,952 |
| | | 11-14 | – | | – |
| NINFEA (Italy, 2005–2016) | Risk group | ≤4 | – | – | 10 |
| | | 5-10 | – | – | |
| | | 11-14 | -0.323 (0.573) | 10 | |
| | Reference group | ≤4 | – | – | 151 |
| | | 5-10 | | | |
| | | 11-14 | 0.028 (0.872) | | 151 |
| | Only-child group | ≤4 | – | – | 571 |
| | | 5-10 | – | | – |
| | | 11-14 | -0.008 (0.988) | | 571 |
| All cohorts | Risk group | ≤4 | 0.038 (0.972) | 307 | 425 |
| | | 5-10 | -0.114 (0.998) | 231 | |
| | | 11-14 | -0.375 (0.823) | 83 | |
| | Reference group | ≤4 | -0.001 (0.976) | 8,481 | 12,315 |
| | | 5-10 | -0.051 (0.950) | | 7,049 |
| | | 11-14 | -0.178 (0.925) | | 2,783 |
| | Only-child group | ≤4 | 0.069 (1.003) | 25,533 | 52,586 |
| | | 5-10 | 0.052 (1.006) | | 37,828 |
| | | 11-14 | -0.029 (0.988) | | 18,535 |

*(Continued)*

**Table 3.** (Continued)

| Internalising scores | Group | Age groups | Mean z-score (SD) | No of Focal children with available scores (by age group and group), n | No of Focal children with available scores (by group), n |
|---|---|---|---|---|---|

Number of focal children with available externalising behaviour measurements, by group and by age group;

Focal children: term-born children (≥ 37 weeks GA) for whom we are analysing scores of internalising and externalising problems;

Groups: risk group (term-born children with preterm born sibling(s)); reference group (term-born children with term-born sibling(s)); only-child group (term-born children without siblings);

Years given in parentheses are years of recruitment, in respective cohort;

In each age group, if a child had multiple scores, we used the latest available;

Abbreviations: MoBa: Norwegian Mother, Father and Child Cohort Study in Norway; DNBC: Danish National Birth Cohort in Denmark; GenR: the Generation R Study; NINFEA: Nascita e INFanzia: gli Effetti dell'Ambiente.

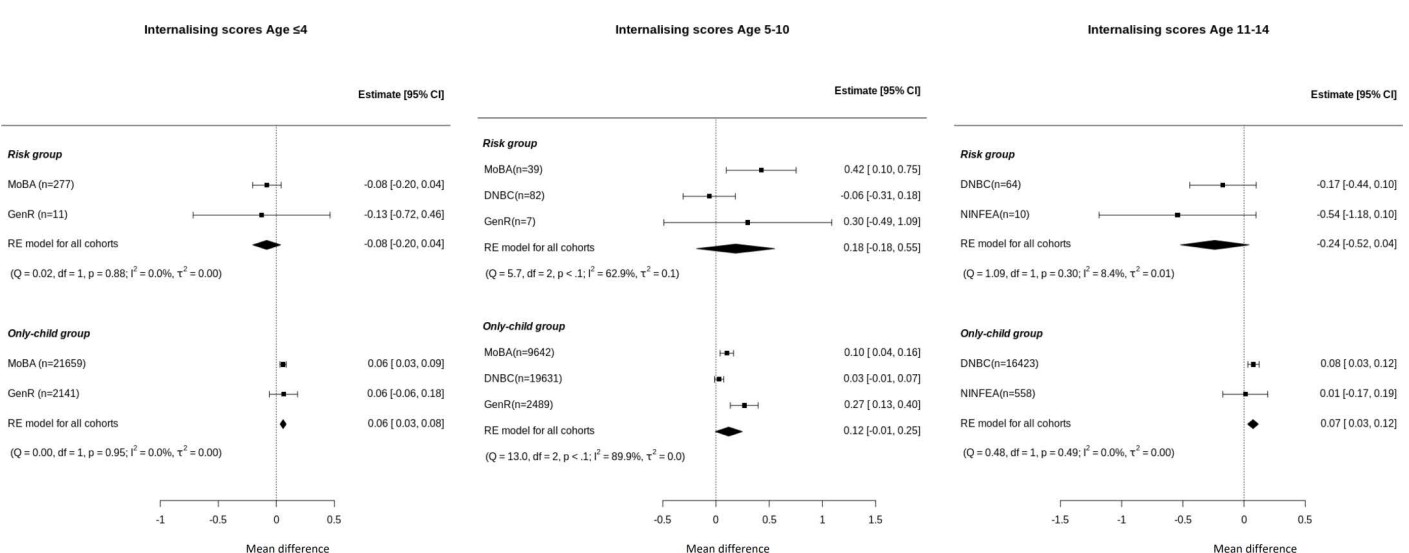

**Fig 2. Forest plot of association between having a preterm sibling and higher or lower internalising scores (z-score) in risk and only-child groups compared to reference group by age group.** Internalising scores (z-score) at ages <=4 years, 5 to 10 and 11 to 14 years using two-stage IPD meta-analysis adjusted models. *Adjusted for age of the mother at birth, mother's education level, pregnancy smoking and alcohol intake during pregnancy. Estimates indicate the difference in standardized internalising scores of children in risk and only-child groups compared to children in the reference group. Positive number means that the group has higher risk compared to reference group and negative number means that group has lower risk than the reference group. In the forest plot: the dot represents the individual adjusted estimates, whiskers cover the 95% CI, n is the number of children included (next to cohort names), and the diamond represents the overall estimates. *I*2 and τ2 statistics are presented under each group.

association between having siblings versus not and externalising behaviour. It also provided evidence supporting that children with an age gap of three years or less with their siblings had poorer externalising outcomes compared to those with higher age gaps. When it came to self-control, the study reported that closely spaced children exhibited better self-control compared to those with wider age gap.

Our results suggest that having a preterm sibling is not in itself a risk factor for increased internalising and externalising behaviour. Yet, children without siblings had higher externalising and internalising scores than those with only term-born siblings. In this study we adjusted for parental education (an indicator of socio-economic status), smoking and alcohol use during pregnancy. These variables have been shown to be associated with preterm birth and with internalising and

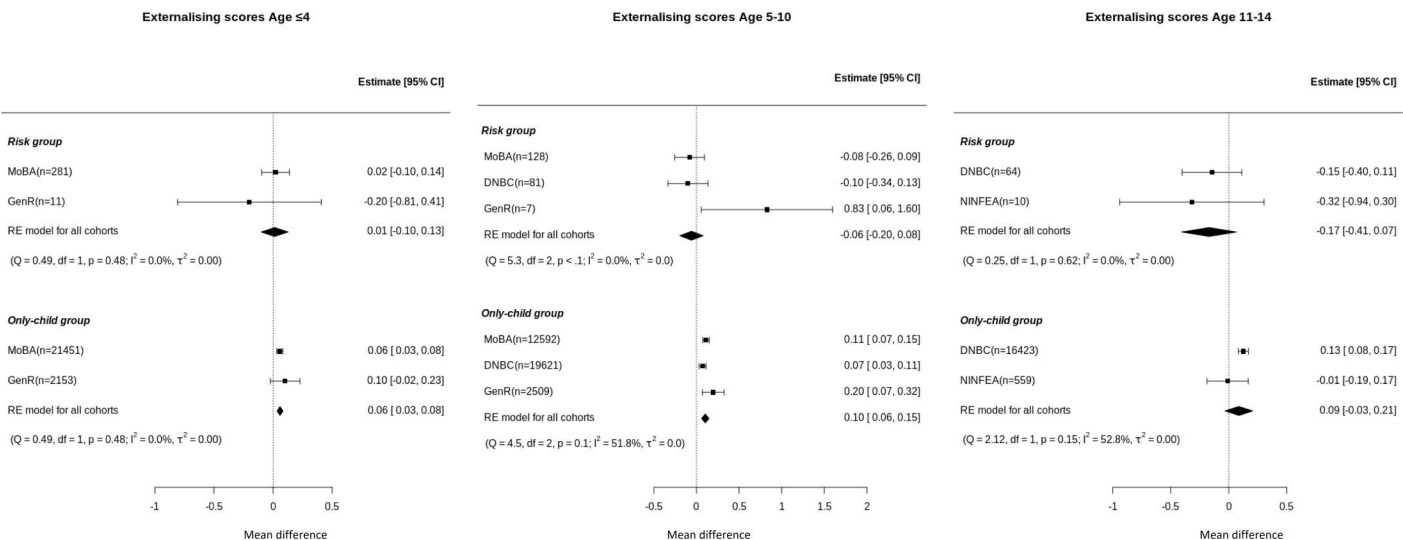

**Fig 3. Forest plot of association between having a preterm sibling and higher or lower externalising scores (z-score) in risk and only-child groups compared to reference by age group.** Externalising scores (z-score) at ages <=4 years, 5 to 10 and 11 to 14 years using two-stage IPD meta-analysis adjusted models. *Adjusted for age of the mother at birth, mother's education level, pregnancy smoking and alcohol intake during pregnancy. Estimates indicate the difference in standardized externalising scores of children in risk and only-child groups compared to children in the reference group. Positive number means that the group has higher risk compared to reference group and negative number means that group has lower risk than the reference group. In the forest plot: the dot represents the individual adjusted estimates, whiskers cover the 95% CI, n is the number of children included (next to cohort names), and the diamond represents the overall estimates. *I2* and τ2 statistics are presented under each group.

externalising behaviour. However, results of unadjusted and adjusted analysis suggest that these confounders did not affect the results observed. However, the increased internalising and externalising behaviour in children without siblings compared to children with term siblings, suggests that other factors not examined in the current study might have contributed to the behavioural results observed. For example, studies investigating mental health outcomes of children whose siblings are affected by autism or chronic diseases have suggested that birth order and the higher externalising in the sibling may play a significant role in the outcome of the focal child [38,39]. In our study, the focal children in the risk group are older than their preterm sibling, and we were able to investigate whether the arrival of the preterm child affected behaviour of the focal child. However, we have not been able to test whether internalising and externalising scores of the term-born children were comparable to scores of their preterm siblings. In addition, as suggested in previous studies, the age gap between siblings might influence the associations, however, in the current study we have not been able to examine the effect of the age difference between the focal children and the preterm children [37]. Equally, we have not been able to investigate the association between the age of the focal child at time of birth of the preterm sibling and the outcomes. Other variables that could explain the internalising and externalising behaviour scores, such as mother's marital status, information on short- or long-term hospitalization and medical complications of the preterm child, the available support network, were not available. Considering all the above, the validity of the results of our study is impacted by the variables available. It has also been shown that the source of the behavioural reports may influence the findings. Most of the above cited studies [17,19,38,39], including the current study, share the study design whereby the outcomes are based on parental reports. Previous research suggests that parents' and children's reports on behaviour tend to vary, with parents more likely to report lower rates of child symptoms than children themselves [40–42]. Factors such as the children's sex and age, type of outcome, socio-economic characteristics and parent-child relationship have been associated with discrepancies between parent and child reports [43]. It has also been suggested that the presence of a sibling in the household is associated with mothers reporting less behavioural problems in their children [42]. The authors of that

paper hypothesized that the presence of other children in the family may raise the threshold for which mothers consider her child's behaviour as problematic [42]. Therefore, it is possible that the higher scores observed in the only-child group could be a result of such response from mothers, biasing maternal reports. Hence, one may also speculate that parents of preterm children, who often have behavioural and neurodevelopmental difficulties, may be biased towards under-reporting the internalising and externalising problems of their term-born children. Such possible biases can only be overcome with data collection from other informants, such as teachers, or methods, such as standardised behaviour observation. Additionally, it is important to consider the interplay between the parents' own health, the family environment and dynamics, and how parents report outcomes of their children. A previous study conducted in the UK supported a within-family interrelations of maternal, paternal and child mental health from early childhood to adolescence [44]. However, that study does not consider family size or structure, nor cumulation or simultaneousness of other significant events [45]. These could be important areas of improvement in future studies.

## Strengths and limitations

Our study has several strengths: we have a large sample size of 65,711 focal children, and their siblings, with harmonised variables including internalising and externalising behavioural measurements validated by CBCL and SDQ. Also, we have been able to test the robustness of our results by conducting two- and one-stage IPD meta-analysis, which yielded comparable results. Another strength is that we have used a federated analysis approach, using DataSHIELD, which enabled us to remotely conduct IPD analysis from four cohorts across Europe, without the need to physically transfer or pool any data.

However, this study also has important limitations. The included cohorts have collected internalising and externalising behaviour information at different ages, leading to inability to include all the cohorts simultaneously across all age groups. Although we have created three age groups and standardised the outcomes to minimise variations caused by the different ages at which data was collected within and across the included cohorts, we observed some heterogeneity in the age groups 5–10 and 11–14. We have not been able to further examine the heterogeneity, for example using meta-regression, due to the small number of cohorts and to the unavailability of cohort-level variables. However, we acknowledge that the considerable heterogeneity observed in internalising behaviour, in the age groups 5–10, could result from the difference in ages at which data was collected in each cohort within this age group. Equally, specific population characteristics could contribute to the differences between studies. Also, there was some risk of misclassification of siblings. For example, mothers included in a cohort may have had additional children that we were not aware of. We ensured that all children in the only-child group were single children at time of entry in the cohort, by excluding all mothers with older children who were not included in the cohort. However, it is possible that there are younger children in all groups who might not have been enrolled in the cohorts. Another limitation is that due to high missingness of maternal psychiatric disorders before and during pregnancy, we have not been able to assess the effect of maternal mental health on the children's behavioural outcomes. Maternal (and the other parent) mental health could confound and/or modify the association presented in this research; but, the magnitude and the directionality of the change upon the reported effects would be hard to estimate from the literature. The parental health status may indeed act upon the exposure (i.e., having a sibling born preterm) and the outcome (i.e., alteration on internalising or externalising signals) via environmental and genetic predispositions. The association on both the exposure and the outcome can be relatively balanced (shared environment confounders) or un-balanced (modifying either the propensity of the exposure to change the outcome and/or the proportion of cases). We recommend that current and future cohort studies strengthen their data collection methods and tracking of maternal mental health variables. We also recommend continuing multi-cohort studies and recommend combining them with other clinical designs to continue triangulating the effects of parental and sibling health upon a child's mental health. Additionally, it must be acknowledged that we could not perform robustness analyses to infer the contribution of family dynamics, parental health or the role of the support systems on later outcomes. These measures were unfortunately missing from

the data of the EU Child cohort network (https://euchildcohortnetwork.eu) and we recommend future harmonization across cohorts to include factors such as adverse childhood experiences (ACEs), parental relationships, schooling and education systems in the databases. We also recommend developing and linking longitudinal measures of maternal, paternal and child health data to further model their dynamics across the life-course.

The risk, reference and only-child group differ considerably in size, with the risk group being formed by 1%, the reference group by 19% and the only-child group by 83% of the focal children with available internalising and externalising measures. First, the small numbers are due to the low percentage of preterm births across all cohorts (n = 13,612, 6%), which closely corresponds to the proportions of preterm births in the respective countries [46–50]. Consequently, the percentage of focal children with preterm born siblings is low. Secondly, due to our exclusion criteria, from the total 13,612 preterm born children in the source populations, only 1,810 (13%) had term-born siblings. Amongst these preterm children, 529 (4%) fulfilled the birth position criterion, of being the preterm sibling born closest after their term-born sibling. Despite our exclusion criteria differing by group, they were specifically set to 1) identify children inside the cohorts with and without siblings, 2) differentiate those born at term with and without preterm born siblings, 3) fulfil a specific birth position. In addition, excluding children without internalising and externalising measurements led to 28% of the original groups to be excluded (19%, 16% and 31% of children excluded in the risk, reference and only-child groups, respectively). Altogether, this potential loss of information, particularly in the risk group, leads to loss of statistical power and increases the uncertainty of the estimates. The method of individual participant meta-analysis was used to help mitigating the risk of Type II error due to small sample size in the risk group. In the two-stage meta-analysis we fitted a fixed-effects linear regression model in each cohort separately and then combined the cohort-specific estimates using random-effects meta-analysis. The one-stage meta-analysis considers all cohorts as one by pooling the data from all cohorts together and thus maximizing their sample size. It is also to be acknowledged how possible participation bias, more specifically potentially higher recruitment rates from mothers with only one child, may have impacted the numbers observed in each group. Yet, we acknowledge that our sample size in the risk group and subsequently the potential loss of reliability of our findings in this group are important limitations. Similarly, the countries included in this study have relatively low percentage of preterm birth, therefore our findings might not be generalizable to other countries with much higher prevalence of preterm birth. We are limited by the small sample size in the risk group and therefore recommend that future studies with a larger sample size will be needed to better understand the observed associations.

Data missingness in several variables is another potential source of bias in this study. The data collection methods used in the four birth cohorts (i.e., telephone interviews, face-to-face interviews, electronic questionnaires) typically generate high risk of missing data. In addition, as indicated above, both CBCL and SDQ questionnaires are parent-reported, which may be at risk of reporting bias. However, a study has suggested that self and parent reported SDQs were equally effective at predicting presence of psychiatric disorders [51]. Yet, future studies would benefit from including multiple sources report. Furthermore, due to size of the risk group, we have been unable to test how the siblings' category of gestational age (i.e., extremely preterm, very preterm, late preterm) affected the outcomes, by stratifying the categories. Finally, this study included only participants living in high-income European countries with good and supportive healthcare and social systems for parents and children, which limits the generalizability of our findings to other populations. Future studies including other regions and countries, where incidence of preterm birth is higher and healthcare systems are weaker, might provide different results to those observed in the current study.

## Conclusions

We found absence of evidence that having a sibling who has been born preterm compared to a term sibling constitutes a risk for increased internalising and externalising behaviour of their older term-born siblings. In contrast, our results showed higher internalising and externalising scores amongst children in the only-child group compared with children who had only term-born siblings. However, these results could, in part, be due to use of parent reported outcomes or to the reduced

sample size in the group of focal children with preterm siblings. Other limitations of this study include our inability to analyse maternal mental health variables and the small sample size of the risk group. Considering this study's limitations, we highlight important areas for improvement in future longitudinal studies, such as inclusion of sibling's mental health, increased sample size of the risk group, inclusion of regions with high prevalence of preterm birth and weaker social and healthcare systems. Understanding whether preterm birth impacts the health of the siblings is of clinical relevance. We recommend replication studies including child, parent, and teacher reported outcomes, highlighting the need to understand the interplay between parents' mental health and reported outcomes, and family structures.

## Supporting information

**S1 Text. Cohorts' information.**
(DOCX)

**S2 Text. Cohort-specific acknowledgments.**
(DOCX)

**S1 Fig. Study Design.**
(TIF)

**S2 Fig. Across all ages analyses.**
(DOCX)

**S1 Table. Descriptive of the groups including children with and without behaviour measurements.**
(DOCX)

**S2 Table. Cohort specific estimates.**
(DOCX)

**S3 Table. One-stage IPD meta-analysis estimates.**
(DOCX)

## Acknowledgments

We are grateful to all the participating families in Norway (MoBa), Denmark (DNBC), the Netherlands (GenR) and Italy (NINFEA). Within each cohort we would also like to thank all the research collaborators, general practitioners, hospitals, midwives, nurses, and pharmacies involved. In addition, we are very grateful to each cohort and team, principal investigators, researchers, and administrators for their ongoing work in these cohorts. Finally, the authors would like to acknowledge the DataSHIELD Community. Please see S2 Text for cohort specific acknowledgments.

## Author contributions

**Conceptualization:** Wnurinham Silva, Eero Kajantie, Sylvain Sebert.

**Data curation:** Wnurinham Silva, Demetris Avraam, Lorenzo Richiardi, Henning Tiemeier.

**Formal analysis:** Wnurinham Silva, Demetris Avraam.

**Funding acquisition:** Wnurinham Silva, Lorenzo Richiardi, Eero Kajantie, Sylvain Sebert.

**Investigation:** Wnurinham Silva.

**Methodology:** Wnurinham Silva, Demetris Avraam, Lorenzo Richiardi, Anne-Marie Nybo Andersen, Eero Kajantie, Sylvain Sebert.

**Project administration:** Wnurinham Silva.

**Resources:** Wnurinham Silva, Eero Kajantie, Sylvain Sebert.

**Software:** Wnurinham Silva, Demetris Avraam.

**Supervision:** Anne-Marie Nybo Andersen, Eero Kajantie, Sylvain Sebert.

**Validation:** Wnurinham Silva, Demetris Avraam, Eero Kajantie, Sylvain Sebert.

**Visualization:** Wnurinham Silva.

**Writing – original draft:** Wnurinham Silva, Demetris Avraam, Eero Kajantie, Sylvain Sebert.

**Writing – review & editing:** Wnurinham Silva, Demetris Avraam, Luise Cederkvist, Johanna Lucia Nader, Maja Popovic, Hanan El Marroun, Jennifer R. Harris, Lorenzo Richiardi, Henning Tiemeier, Timothy James Cadman, Julia Jaekel, Anne-Marie Nybo Andersen, Eero Kajantie, Sylvain Sebert.

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
