## [Decision Letter · Decision Letter 0]

10 Jul 2024

PMEN-D-24-00079

Internalising and externalising behaviour in siblings of children born preterm

PLOS Mental Health

Dear Dr. Silva,

Thank you for submitting your manuscript to PLOS Mental Health. After careful consideration, we feel that it has merit but does not fully meet PLOS Mental Health’s publication criteria as it currently stands. Therefore, we invite you to submit a revised version of the manuscript that addresses the points raised during the review process.

The manuscript has been evaluated by three reviewers, and their comments are available below.

The reviewers have raised a number of concerns that need attention. They request additional information on methodological aspects of the study (such as the inclusion of information on the sample size and response rate), revisions to the statistical analyses and they question the internal and external validity of the results reported.

Could you please revise the manuscript to carefully address the concerns raised?

We look forward to receiving your revised manuscript.

Kind regards,

Avanti Dey, PhD

Staff Editor

PLOS Mental Health

Journal Requirements:

Additional Editor Comments (if provided):

Reviewers' comments:

Reviewer's Responses to Questions

**Comments to the Author**

1. Does this manuscript meet PLOS Mental Health’s publication criteria ? Is the manuscript technically sound, and do the data support the conclusions? The manuscript must describe methodologically and ethically rigorous research with conclusions that are appropriately drawn based on the data presented.

Reviewer #1: Partly

Reviewer #2: Yes

Reviewer #3: Yes

2. Has the statistical analysis been performed appropriately and rigorously?

Reviewer #1: Yes

Reviewer #2: Yes

Reviewer #3: Yes

3. Have the authors made all data underlying the findings in their manuscript fully available (please refer to the Data Availability Statement at the start of the manuscript PDF file)?

Reviewer #1: Yes

Reviewer #2: Yes

Reviewer #3: Yes

4. Is the manuscript presented in an intelligible fashion and written in standard English?

Reviewer #1: Yes

Reviewer #2: Yes

Reviewer #3: Yes

5. Review Comments to the Author

Reviewer #1: General: The authors mix present and past tenses throughout the manuscript (e.g., lines 167 – 172). Please edit and be consistent.

Lines 93-95: What are the short- and long-term impacts of preterm birth on parental stress and depression/mental health? More details are needed.

Lines 189-191: The term “null parity” needs to be explained.

Figure 2: Unnecessary as the details are explained in the manuscript. I suggest putting it into the supplement.

Lines 206-211: CBCL and MFQ should be explained in more detail: what do they measure, which (sub)scales were used, etc. Also, details for the calculation of the z scores need to be provided.

Lines 222-226: Details are needed how maternal education was divided into high, medium, and low (e.g., based on which school degree, school years, or similar). Also, why was maternal mental health not considered as a confounder? Research has shown that maternal mental health has significant impact on children's mental health and behavior.

Lines 246-249: This approach is unclear: why was it done? What exactly was done? More details are needed.

Table 2: The Sex NA row is not necessary as it has no entries. Focal birth weight is listed but not explained why it is relevant and what the norms for birth weight are.

Lines 322-356: Meta-analysis results are available in the figure but missing in the text! Please include the results (MD, 95% CI, p) in the text to avoid confusion. High heterogeneity was found for some results, but no analysis was provided to further examine it.

Figures 2-4: Captions are too long, consider shortening them. Further, it needs to be explained what the positive and negative estimates in the forest plots represent. Does a positive estimate mean “no difference between the groups”? Renaming the x axis title “adjusted estimate” to something that better explains the outcome could help avoiding this problem.

Lines 376-377: Increased or decreased scores? The children had scores but how did they differ (or not differ) in their scores compared to the reference group?

Lines 382-384: Differences (or the lack thereof) in what? The authors explained how the results of this study compare to the present literature but failed to provide a takeaway message (does this study add to the current literature or contradict it? What should be the next step?).

Line 389: Avoid using the term "only-child children" as it can lead to confusion. Instead you could use "children without siblings" or "only-children", as done before in the manuscript.

Line 392: Similarly, avoid using the term "not only-child peers". It could be replaced by a simpler version like "children with siblings" or "peers with siblings", as done before.

Line 394: The authors are encouraged to describe the mentioned study (39) in more detail as it will add important information to the discussion.

Line 395: The results of this study do not suggest that children’s behaviour is affected by a multitude of factors. In detail, this study showed that children with pre-term siblings do not differ in their internalizing or externalizing behavior compared to children with term siblings, though children without siblings show increased internalizing and externalizing behavior compared to children with term siblings. Other factors that might have contributed to the behavioral results were not examined. Please revise this statement to accurately reflect the results.

Lines 405-427: The source of behavioral reporting is indeed an important factor that needs to be considered when looking at the results. The authors could show that parents with more than one child have a higher threshold of categorizing their child’s behavior as problematic. However, the authors than argue that parents with only one child will overreport their child’s behavioral problems. While this is an interesting thought, providing a new direction for future research, it should not be mentioned in the abstract (lines 76-77) as this has not been investigated yet.

Line 438-440: Heterogeneity was not investigated in this study, though it should be to examine possible factors that could have led to high heterogeneity. In typical meta-analyses, this is done via meta-regression.

Line 445-448: This is an important point that also needs to be mentioned in the methods section (see comments about confounders above).

Lines 463-465: This is indeed a serious limitation, which also needs to be mentioned and discussed in more detail in the discussion section. Would the results change if the group sizes change (e.g., would the significant differences between the only-child and reference groups become non-significant)?

Lines 475-478: This is a great suggestion: research focus should shift toward regions with high prevalence of preterm birth. Differences in demographics, culture, health care system, etc. could affect behavioral outcomes of children with preterm siblings! The cohorts used for this study were indeed all conducted in high-income countries with excellent health care for parents and children – could this have affected the results?

Lines 488-489: Avoid using the term "mental health problems", use the previously used term “internalising and externalising behaviour” instead to keep the manuscript consistent.

Lines 489-492: Various suggested factors from the discussion section are missing here (e.g., siblings mental health, sample size of risk group, regions with high prevalence of preterm birth) but should be included.

Supplement S3, S4, S6: not mentioned in the text, only in the supplement section (706-712). Please revise the text to include them.

S4: Not found in supplement.

S6 ans S10: The text titles and supplement titles are different. Please revise.

Reviewer #2: This study was based on a database of harmonised data from

four birth cohorts participating in the European Child Cohort Network. A total of 65,711

term-born children (49% girls) across the four cohorts were grouped as follows: risk

group (with younger preterm born siblings; n=427), reference group (with only term

siblings; n=12,371); and only-child group (without siblings; n=52,913) to assess differences in internalising and externalising behaviours in siblings as reported by parents on the SDQ or CBCL.

Abstract- well written, however suggest authors reframe conclusion statement re “This finding is consistent with previous reports suggesting overreporting

of behavioural problems by parents of an only-child.” as there may be other reasons for over reporting of behavioural problems in one child only parents.

Introduction- succinct

Methods- several limitations but my main concern is comparing cohorts using two different scales ie SDQ data versus CBCL- can authors please comment on reliability and validity of methods used

Results - cohorts are stratified into age groups- Do we know age at which the focal child had sibling and temporal association between the event and the assessment timings

discussion- authors do discuss the limitations.

Conclusion- fair

Whilst there are multiple limitations, results do contribute to the limited evidence on the subject.

Reviewer #3: • The authors state that “it has been found that family stress levels remain elevated seven years after the birth of a preterm child, (11). Thus, it can be hypothesized that siblings may also be affected by having a preterm born sibling”. Can authors add a sentence or two to describe how elevated family stress would impact the children?

• The study has not examined several important confounding factors that could have influenced the externalizing and internalizing symptoms. These include

o socio-economic status – families from lower SES will have fewer resources to mange the needs of a pre-term child, which may lead to increased family stress and behavioural issues in the older sibling

o parents’ marital status – single mothers will have greater difficulty in providing care to the older sibling while caring for a pre-term child

o age of the focal child at the birth of the preterm sibling – children who were older at the time of the birth of the pre-term sibling may have better coping skills to adjust to the demands of having a pre-term sibling

o whether the preterm sibling required long term hospitalized care or had medical needs that required frequent hospitalizations– if the mother had to accompany the pre-term sibling in hospital for long periods, this would have greatly impacted the mental health of the sibling

o family support – family support (e.g., from extended family) may influence how the mother coped with the needs of the preterm child while caring for the older sibling.

A separate paragraph/paragraphs needs to be added to the discussing these confounding factors, reflecting on previous literature and how they may have influenced the results of the study. The limitations section should clearly indicate that the validity of the results is impacted by these confounding factors.

6. PLOS authors have the option to publish the peer review history of their article (what does this mean? ). If published, this will include your full peer review and any attached files.

**Do you want your identity to be public for this peer review?** For information about this choice, including consent withdrawal, please see our Privacy Policy .

Reviewer #1: No

Reviewer #2: No

Reviewer #3: No

---

## [Decision Letter · Decision Letter 1]

26 Dec 2024

PMEN-D-24-00079R1

Internalising and externalising behaviour in siblings of children born preterm

PLOS Mental Health

Dear Dr. Sebert,

Thank you for submitting your manuscript to PLOS Mental Health. After careful consideration, we feel that it has merit but does not fully meet PLOS Mental Health’s publication criteria as it currently stands. Therefore, we invite you to submit a revised version of the manuscript that addresses the points raised during the review process.

We look forward to receiving your revised manuscript.

Kind regards,

Gellan Karamallah Ramadan Ahmed

Academic Editor

PLOS Mental Health

Journal Requirements:

Additional Editor Comments (if provided):

Reviewers' comments:

Reviewer's Responses to Questions

**Comments to the Author**

1. If the authors have adequately addressed your comments raised in a previous round of review and you feel that this manuscript is now acceptable for publication, you may indicate that here to bypass the “Comments to the Author” section, enter your conflict of interest statement in the “Confidential to Editor” section, and submit your "Accept" recommendation.

Reviewer #1: All comments have been addressed

Reviewer #4: (No Response)

2. Does this manuscript meet PLOS Mental Health’s publication criteria ? Is the manuscript technically sound, and do the data support the conclusions? The manuscript must describe methodologically and ethically rigorous research with conclusions that are appropriately drawn based on the data presented.

Reviewer #1: Yes

Reviewer #4: Yes

3. Has the statistical analysis been performed appropriately and rigorously?

Reviewer #1: Yes

Reviewer #4: Yes

4. Have the authors made all data underlying the findings in their manuscript fully available (please refer to the Data Availability Statement at the start of the manuscript PDF file)?

Reviewer #1: Yes

Reviewer #4: Yes

5. Is the manuscript presented in an intelligible fashion and written in standard English?

Reviewer #1: Yes

Reviewer #4: Yes

6. Review Comments to the Author

Reviewer #1: Thank you for revising your manuscript according to the previous suggestions. I have no further comments, other than to accept this important study for publication.

Reviewer #4: This study "Internalising and externalising behaviour in siblings of children born preterm" examined whether term-born children with preterm siblings (risk group) exhibit higher internalizing and externalizing behavior problems compared to term-born children with term-born siblings (reference group) and children without siblings (only-child group). Data from 65,711 term-born children across four European cohorts were analyzed using z-standardized behavior scores based on parental reports. No significant differences were found between the risk and reference groups across all age ranges (≤4, 5–10, and 11–14 years). However, children without siblings showed higher internalizing and externalizing behavior scores compared to those with term-born siblings. A limitation was the small sample size of the risk group, reducing statistical power.

Here are my comments:

Abstract

1. The abstract can be made more concise by focusing on the key findings and their implications, while omitting less critical details (e.g., specific cohort statistics or methods not central to the conclusion).

2. Clarify the hypothesis and ensure it is aligned with the results and conclusion.

Introduction

1. Provide a more in-depth explanation of why sibling dynamics in the context of preterm births are significant for mental health research.

2. Expand the discussion of existing studies, particularly those with conflicting findings, to better justify the study's hypothesis.

3. Clearly articulate the research hypothesis and its rationale, ensuring alignment with the study design and outcomes.

Methods

1. While the harmonization across cohorts is a strength, provide additional details on how differences in data collection (e.g., timing, instruments) were addressed.

2. Justify the exclusion of maternal mental health data due to missingness and discuss its potential impact on findings.

3. Elaborate on how heterogeneity between cohorts was managed, especially in the meta-analysis.

4. Highlight steps taken to mitigate the effects of small sample sizes in the risk group.

Discussion

1. Strengthen comparisons with existing studies, particularly in explaining discrepancies.

2. Discuss potential biases (e.g., parental reporting) and their impact on the findings more comprehensively.

3. Expand on the implications of the findings for clinical practice and policy, especially regarding interventions for siblings of preterm children.

4. Provide concrete suggestions for addressing identified gaps, such as collecting direct sibling-reported data or including larger sample sizes.

Conclusion

1. Summarize the findings and their implications succinctly, avoiding repetition from other sections.

2. Briefly reiterate the key limitations to provide context to the conclusion.

Recommendations: End with clear recommendations or calls to action for future studies or policy changes.

7. PLOS authors have the option to publish the peer review history of their article (what does this mean? ). If published, this will include your full peer review and any attached files.

**Do you want your identity to be public for this peer review?** For information about this choice, including consent withdrawal, please see our Privacy Policy .

Reviewer #1: No

Reviewer #4: **Yes: ** Ahmad Neyazi

---

## [Decision Letter · Decision Letter 2]

11 Mar 2025

PMEN-D-24-00079R2

Internalising and externalising behaviour in siblings of children born preterm

PLOS Mental Health

Dear Dr. Sebert,

Thank you for submitting your manuscript to PLOS Mental Health. After careful consideration, we feel that it has merit but does not fully meet PLOS Mental Health’s publication criteria as it currently stands. Therefore, we invite you to submit a revised version of the manuscript that addresses the points raised during the review process.

We look forward to receiving your revised manuscript.

Kind regards,

Gellan Karamallah Ramadan Ahmed

Academic Editor

PLOS Mental Health

Journal Requirements:

Additional Editor Comments (if provided):

Reviewers' comments:

Reviewer's Responses to Questions

**Comments to the Author**

1. If the authors have adequately addressed your comments raised in a previous round of review and you feel that this manuscript is now acceptable for publication, you may indicate that here to bypass the “Comments to the Author” section, enter your conflict of interest statement in the “Confidential to Editor” section, and submit your "Accept" recommendation.

Reviewer #5: (No Response)

2. Does this manuscript meet PLOS Mental Health’s publication criteria ? Is the manuscript technically sound, and do the data support the conclusions? The manuscript must describe methodologically and ethically rigorous research with conclusions that are appropriately drawn based on the data presented.

Reviewer #5: Yes

3. Has the statistical analysis been performed appropriately and rigorously?

Reviewer #5: Yes

4. Have the authors made all data underlying the findings in their manuscript fully available (please refer to the Data Availability Statement at the start of the manuscript PDF file)?

Reviewer #5: No

5. Is the manuscript presented in an intelligible fashion and written in standard English?

Reviewer #5: Yes

6. Review Comments to the Author

Reviewer #5: As a new reviewer to comment this re-revised manuscript, I do commend authors doing good job to address fundamental concerns raised by previous peer reviewers. Personally speaking, this re-revision is indeed better than original one by following previous comments all the reviewers have already shared. Collectively, this study is robust with a large sample size and well-conducted meta-analysis, and has been improved well in the current version. Thus, I remain limited yet essential comments to improve this study, especially related to statistical power, cohort variability, and missing data, to release revision burdens for authors as far as possible:

- The small number of children in the risk group (with preterm siblings) reduces the statistical power, which may limit the reliability of the results for this group. Acknowledging this limitation more explicitly and discuss how it may affect the generalizability of the findings. Future studies could aim to include a larger sample of children with preterm siblings.

- The different ages at which internalising and externalising scores were collected across cohorts may introduce heterogeneity and biases in results. Considering providing more detailed explanations of how cohort-specific differences in measurement age were handled and discuss how this could influence the findings. Cohort-specific analyses may also add clarity.

- The lack of maternal mental health data (a known confounder) due to high missingness could bias the results. It would be helpful to address how the missing maternal mental health data might impact the results and whether future research could include better tracking of this variable.

- This manuscript focuses on sibling status but does not delve deeply into other factors, such as family dynamics, parental health, or support systems, which could also contribute to behavioural outcomes. Those robustness analyses should be added.

- The manuscript uses complete case analysis but does not provide sufficient detail on how missing data was handled across the cohorts.

- Study heterogeneity should be examined for the between-cohort condition. The manuscript reports pooled results without sufficiently discussing how the findings may vary across different cohorts.

- A significant confounder - the potential bias from parental reporting of behaviors, should be considered. Parental reports may be biased, especially in families with only-children, where parents might report more behavioural issues.

- I am not sure whether this data contain information for the potential influence of the age gap between siblings on the behavioural outcomes. If gathering this data, this age gap should be modeled into the statistical analysis, given it has well-acknowledged for roles in the within-family mental health.

7. PLOS authors have the option to publish the peer review history of their article (what does this mean? ). If published, this will include your full peer review and any attached files.

**Do you want your identity to be public for this peer review?** For information about this choice, including consent withdrawal, please see our Privacy Policy .

Reviewer #5: No

---

## [Decision Letter · Decision Letter 3]

25 Apr 2025

Internalising and externalising behaviour in siblings of children born preterm

PMEN-D-24-00079R3

Dear Dr Sebert,

We are pleased to inform you that your manuscript 'Internalising and externalising behaviour in siblings of children born preterm' has been provisionally accepted for publication in PLOS Mental Health.

Best regards,

Gellan Karamallah Ramadan Ahmed

Academic Editor

PLOS Mental Health

Reviewer Comments (if any, and for reference):

Reviewer's Responses to Questions

**Comments to the Author**

1. If the authors have adequately addressed your comments raised in a previous round of review and you feel that this manuscript is now acceptable for publication, you may indicate that here to bypass the “Comments to the Author” section, enter your conflict of interest statement in the “Confidential to Editor” section, and submit your "Accept" recommendation.

Reviewer #5: All comments have been addressed

2. Does this manuscript meet PLOS Mental Health’s publication criteria ? Is the manuscript technically sound, and do the data support the conclusions? The manuscript must describe methodologically and ethically rigorous research with conclusions that are appropriately drawn based on the data presented.

Reviewer #5: Yes

3. Has the statistical analysis been performed appropriately and rigorously?

Reviewer #5: Yes

4. Have the authors made all data underlying the findings in their manuscript fully available (please refer to the Data Availability Statement at the start of the manuscript PDF file)?

Reviewer #5: Yes

5. Is the manuscript presented in an intelligible fashion and written in standard English?

Reviewer #5: Yes

6. Review Comments to the Author

Reviewer #5: Thank to authors for addressing all my concerns. I have no more ones.

7. PLOS authors have the option to publish the peer review history of their article (what does this mean? ). If published, this will include your full peer review and any attached files.

**Do you want your identity to be public for this peer review?** For information about this choice, including consent withdrawal, please see our Privacy Policy .

Reviewer #5: No
